# Discovery of Useful Questions as Auxiliary Tasks

Vivek Veeriah[1]          Matteo Hessel[2]          Zhongwen Xu[2]          Richard Lewis[1]

Janarthanan Rajendran[1]          Junhyuk Oh[2]          Hado van Hasselt[2]          David Silver[2]

Satinder Singh[1,2]

## Abstract

Arguably, intelligent agents ought to be able to *discover* their own questions so that in learning answers for them they learn unanticipated useful knowledge and skills; this departs from the focus in much of machine learning on agents learning answers to externally defined questions. We present a novel method for a reinforcement learning (RL) agent to discover questions formulated as general value functions or GVFs, a fairly rich form of knowledge representation. Specifically, our method uses non-myopic meta-gradients to learn GVF-questions such that learning answers to them, as an auxiliary task, induces useful representations for the main task faced by the RL agent. We demonstrate that auxiliary tasks based on the discovered GVFs are sufficient, on their own, to build representations that support main task learning, and that they do so better than popular hand-designed auxiliary tasks from the literature. Furthermore, we show, in the context of Atari 2600 videogames, how such auxiliary tasks, meta-learned alongside the main task, can improve the data efficiency of an actor-critic agent.

An increasingly important component of recent approaches to developing flexible, autonomous agents is posing useful *questions* about the future for the agent to learn to answer from experience. The questions can take many forms and serve many purposes. The answers to prediction or control questions about suitable features of states may directly form useful representations of state (Singh et al., 2004). Alternatively, prediction and control questions may define *auxiliary tasks*, that drive representation learning in the aid of a main task (Jaderberg et al., 2017). Goal-conditional questions may also drive the acquisition of a diverse set of skills, even before the main task is known, forming a basis for policy composition or exploration (Andrychowicz et al., 2016; Veeriah et al., 2018; Eysenbach et al., 2018; Florensa et al., 2018; Mankowitz et al., 2018; Riedmiller et al., 2018).

In this paper, we consider questions in the form of *general value functions* (GVFs, Sutton et al., 2011), with the purpose of using the discovered GVFs as auxiliary tasks to aid the learning of a main reinforcement learning (RL) task. We chose the GVF formulation for its flexibility: according to the *reward hypothesis* (Sutton & Barto, 2018), any goal might be formulated in terms of a scalar signal, or *cumulant* (White, 2015), whose discounted sum must be maximized. Additionally, GVF-based auxiliary tasks have been shown in previous work to improve the sample efficiency of reinforcement learning agents engaged in learning complex tasks (Mirowski et al., 2017; Jaderberg et al., 2017).

In the literature, GVF-based auxiliary tasks typically required an agent to estimate discounted sums of suitable handcrafted functions of state, *cumulants* in the GVF terminology, under handcrafted discount factors. It was then shown that by combining gradients from learning the auxiliary GVFs with

the updates from the main task, it was possible to accelerate representation learning and improve performance. It fell, however, onto the algorithm designer to design questions that were useful for the specific task. This is a limitation because not all questions are equally well aligned with the main task (Bellemare et al., 2019), and whether this is the case may be hard to predict in advance.

The paper makes three contributions. First, we propose a principled general method for the automated discovery of questions in the form of GVFs, for use as auxiliary tasks. The main idea is to use meta-gradient RL to discover the questions so that answering them maximises the usefulness of the induced representation on the main task. This removes the need to hand-design auxiliary tasks that are matched to the environment or agent. Our second contribution is to empirically demonstrate the success of non-myopic meta-gradient RL in large, challenging domains as opposed to the approximate and myopic meta-gradient methods from previous work (Xu et al., 2018; Zheng et al., 2018); the non-myopic calculation of meta-gradient proved essential to successfully learn useful questions and should be applicable more broadly to other applications of meta-gradients. Finally, we demonstrate in the context of Atari 2600 videogames that such discovery of auxiliary tasks can improve the data efficiency of an actor-critic agent, when these are meta-learned along side the main task.

# 1 Background

**Brief background on GVFs:** Standard value functions in RL define a question and its answer; the question is "*what is the discounted sum of future rewards under some policy?*" and the answer is the approximate value function. Generalized value functions, or GVFs, generalize the standard value function to allow for arbitrary *cumulant* functions of states in place of rewards, and are specified by the combination of such a cumulant function with a discount factor and a policy. This generalization of standard value functions allows GVFs to express quite general predictive knowledge and, notably, temporal-difference (TD) methods for learning value functions can be extended to learn the predictions/answers of GVFs. We refer to Sutton et al. (2011) for additional details.

**Prior work on auxiliary tasks in RL:** Jaderberg et al. (2017) explored extensively the potential, for RL agents, of jointly learning the representation used for solving the main task and a number of GVF-based auxiliary tasks, such as pixel-control and feature-control tasks based on controlling changes in pixel intensities and feature activations; this class of auxiliary tasks was also used in the multi-task setting by Hessel et al. (2019a). Other recent examples of auxiliary tasks include depth and loop closure classification (Mirowski et al., 2017), observation reconstruction, reward prediction, inverse dynamics prediction (Shelhamer et al., 2017), and many-goals learning (Veeriah et al., 2018). A geometrical perspective on auxiliary tasks was introduced by Bellemare et al. (2019).

**Prior work on meta-learning:** Recently, there has been a lot of interest in exploring *meta-learning* or *learning to learn*. A meta-learner progressively improves the learning process of a learner (Schmidhuber et al., 1996; Thrun & Pratt, 1998) that is attempting to solve some task. Recent work on meta-learning includes learning good policy initializations that can be quickly adapted to new tasks (Finn et al., 2017; Al-Shedivat et al., 2018), improving few-shot learning performance (Mishra et al., 2018; Duan et al., 2017; Snell et al., 2017), learning to explore (Stadie et al., 2018), unsupervised learning (Gupta et al., 2018; Hsu et al., 2018), few-shot model adaptation (Nagabandi et al., 2018), and improving the optimizers (Andrychowicz et al., 2016; Li & Malik, 2017; Ravi & Larochelle, 2017; Wichrowska et al., 2017; Chen et al., 2016; Gupta et al., 2018).

**Prior work on meta-gradients:** Xu et al. (2018) formalized *meta-gradients*, a form of meta-learning where the meta-learner is trained via gradients through the effect of the meta-parameters on a learner also trained via gradients. In contrast to much work in meta-learning that focuses on multi-task learning, Xu et al. (2018) formalized the use of meta-gradients in a way that is applicable also to the single task setting, although not limited to it. They illustrated their approach by using meta-gradients to adapt both the discount factor $\gamma$ and the bootstrapping factor $\lambda$ of a reinforcement learning agent, substantially improving performance of an actor-critic agent on many Atari games. Concurrently, Zheng et al. (2018) used meta-gradients to learn intrinsic rewards, demonstrating that maximizing a sum of extrinsic and intrinsic rewards could improve an agent's performance on a number of Atari games and MuJoCo tasks. Xu et al. (2018) discussed the possibility of computing meta-gradients in a non-myopic manner, but their proposed algorithm, as that of Zheng et al. (2018), introduced a severe approximation and only measured the immediate consequences of an update.

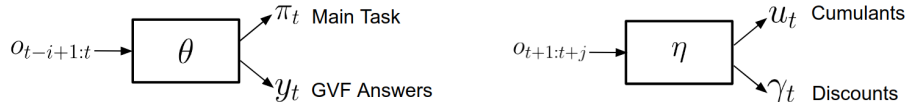

Figure 1: An architecture for discovery: On the left, the **main task** and **answer** network with parameters $\theta$; it takes past observations as input and parameterises (directly or indirectly) a policy $\pi$ as well as the answers to the GVF questions. On the right, the **question** network with parameters $\eta$; it takes future observations as input and parameterises the cumulants and discounts that specify the GVFs.

## 2   The discovery of useful questions

In this section we present a neural network architecture and a principled meta-gradient algorithm for the discovery of GVF-based questions for use as auxiliary tasks in the context of deep RL agents.

### 2.1   A neural network architecture for discovery

The neural network architecture we consider features two networks: the first, on the left in Figure 1, takes the last $i$ observations $o_{t-i+1:t}$ as inputs, and parameterises (directly or indirectly) a policy $\pi$ for the main reinforcement learning task, together with GVF-predictions for a number of discovered cumulants and discounts. We use $\theta$ to denote the parameters of this first network. The second network, referred to as the *question network*, is depicted on the right in Figure 1. It takes as inputs $j$ *future* observations $o_{t+1:t+j}$ and, through the meta-parameters $\eta$, computes the values of *a set* of cumulants $u_t$ and their corresponding discounts $\gamma_t$ (both $u_t$ and $\gamma_t$ are therefore vectors).

The use of future observations $o_{t+1:t+j}$ as inputs to the question network requires us to wait $j$ steps to unfold before computing the cumulants and discounts; this is acceptable because the question and answer networks are only used during training, and neither is needed for action selection. As discussed in Section 1, a GVF-question is specified by a cumulant function, a discount function *and* a policy. In our method, the question network only explicitly parameterises discounts and cumulants because we consider on-policy GVFs, and therefore the policy will always be, implicitly, the latest main-task policy $\pi$. Note however, that since each cumulant is a function of future observations, which are influenced by the actions chosen by the main task policy, the cumulant and discount functions are non-stationary, not just because we are learning the question network parameters, but also because the main-task policy itself is changing as learning progresses.

Previous work on auxiliary tasks in reinforcement learning may be interpreted as just using the network on the left, as the cumulant functions were handcrafted and did not have any (meta-)learnable parameters; the availability of a separate "question network" is a critical component of our approach to discovery, as it enables the agent to discover from experience the most suitable questions about the future to be used as auxiliary tasks. The terminology of question and answer networks is derived from work on TD networks (Sutton & Tanner, 2005). See Makino & Takagi (2008) and Schlegel et al. (2018) for related work on incremental discovery of the structure of TD-networks and GVF-networks (work that does not, however, use meta-gradients and was applied only to relatively simple domains).

### 2.2   Multi-step meta-gradients

In their most abstract form, reinforcement learning algorithms can be described by an update procedure $\Delta\theta_t$ that modifies, on each step $t$, the agent's parameters $\theta_t$. The central idea of meta-gradient RL is to parameterise the update $\Delta\theta_t(\eta)$ by meta-parameters $\eta$. We may then consider the consequences of changing $\eta$ on the $\eta$-parameterised update rule by measuring the subsequent performance of the agent, in terms of a "meta-loss" function $m(\theta_{t+k})$. Such meta-loss may be evaluated after one update (myopic) or $k > 1$ updates (non-myopic). The meta-gradient is then, by the chain rule,

$$\frac{\partial m(\theta_{t+k})}{\partial \eta} = \frac{\partial m(\theta_{t+k})}{\partial \theta_{t+k}} \frac{\partial \theta_{t+k}}{\partial \eta}. \tag{1}$$

Implicit in Equation 1 is that changing the meta-parameters $\eta$ at one time step affects not just the immediate update to $\theta$ on the next time step, but at all future updates. This makes the meta-gradient

---

**Algorithm 1** Multi-Step Meta-Gradient Discovery of Questions for Auxiliary Tasks

---
Initialize parameters $\theta, \eta$
**for** $t = 1, 2, \cdots, N$ **do**
    $\theta_{t,0} \leftarrow \theta_t$
    **for** $k = 1, 2, \cdots, L$ **do**
        Generate experience using parameters $\theta_{t,k-1}$
        $\theta_{t,k} \leftarrow \theta_{t,k-1} - \alpha' \nabla_{\theta_{t,k-1}} \mathcal{L}^{RL}(\theta_{t,k-1}) - \alpha' \nabla_{\theta_{t,k-1}} \mathcal{L}^{ans}(\theta_{t,k-1})$
    **end for**
    $\eta_{t+1} \leftarrow \eta_t - \alpha \nabla_\eta \sum_{k=1}^{L} \mathcal{L}^{RL}(\theta_{t,k})$
    $\theta_{t+1} \leftarrow \theta_{t,L}$
**end for**

---

challenging to compute. A straightforward but effective way to capture the multi-step effects of changing $\eta$ is to build a computational graph which consists of a sequence of updates made to the parameters $\theta$, $\theta_t \rightarrow ... \rightarrow \theta_{t+k}$ with $\eta$ held fixed, ending with a meta-loss evaluation $m(\theta_{t+k})$. The meta-gradient $\frac{\partial m(\theta_{t+k})}{\partial \eta}$ may be efficiently computed from this graph through backward-mode autodifferentiation; this has a computational cost similar to that of the forward computation (Griewank & Walther, 2008), but it requires storage of $k$ copies of the parameters $\theta_{t:t+k}$, thus increasing the memory footprint. We emphasize that this approach is in contrast to the myopic meta-gradient used in previous work, that either ignores effects past the first time step, or makes severe approximations.

### 2.3 A multi-step meta-gradient algorithm for discovery

We apply the meta-gradient algorithm, as presented in Section 2.2, to the discovery of GVF-based auxiliary tasks represented as in the neural network architecture from Section 2.1. The complete pseudo code for the proposed approach to discovery is outlined in Algorithm 1.

On each iteration $t$ of the algorithm, in an inner loop we apply $L$ updates to the agent parameters $\theta$, which parameterise the main-task policy and the GVF answers, using separate samples of experience in an environment. Then, in the outer loop, we apply a single update to the meta-parameters $\eta$ (the question network that parameterises cumulant and discount functions that define the GVFs), based on the effect of the updates to $\theta$ on the meta-loss; next, we make each of these steps explicit.

The inner update includes two components: the first is a canonical deep reinforcement learning update using loss denoted $\mathcal{L}^{RL}$ for optimizing the main-task policy $\pi_t$, either directly (as in policy-based algorithms, e.g., Williams (1992)) or indirectly (as in value-based algorithms, e.g., Watkins (1989)). The second component is an update rule for estimating the answers to GVF-based questions. With slight abuse of notation, we can then denote each inner-loop update as the following gradient descent steps on the pseudo losses denoted with $\mathcal{L}^{RL}$ and $\mathcal{L}^{ans}$:

$$\theta_{t,k} \leftarrow \theta_{t,k-1} - \alpha' \nabla_{\theta_{t,k-1}} \mathcal{L}^{RL}(\theta_{t,k-1}) - \alpha' \nabla_{\theta_{t,k-1}} \mathcal{L}^{ans}(\theta_{t,k-1}). \tag{2}$$

The meta loss $m$ is the sum of the RL pseudo losses associated with the main task updates, as computed on the batches generated in the inner loop; it is a function of meta-parameters $\eta$ through the updates to the answers. We can therefore compute the update to the meta-parameters

$$\eta_{t+1} \leftarrow \eta_t - \alpha \nabla_\eta \sum_{k=1}^{L} \mathcal{L}^{RL}(\theta_{t,k}). \tag{3}$$

This meta-gradient procedure optimizes the *area under the curve* over the temporal span defined by the inner unroll length $L$. Alternatively, the meta-loss may be evaluated on the last *batch* alone, to optimize for final performance. Unless we specify otherwise, we use the area under the curve.

### 2.4 An actor critic agent with discovery of questions for auxiliary tasks

In this section we describe a concrete instantiation of the algorithm in the context of an actor-critic reinforcement learning agent. The network on the left of Figure 1 is composed of three modules: 1) an *encoder* network that, takes the last $i$ observations $o_{t-i+1:t}$ as inputs, and outputs a state representation $x_t$; 2) a *main task* network that, given the state $x_t$ estimates both the policy $\pi$ and a

state value function $v$ (Sutton, 1988) 3) an *answer* network that, given the state $x_t$ approximates the GVF answers. In this paper, functions $\pi$, $v$ and $y$ will be linear functions of state $x_t$.

The *main-task network* parameters $\{\theta^{main}\}$ are only affected by the RL component of update defined in Equation 2. In an actor-critic agent, $\theta^{main}$ is the union of the parameters $\theta^v$ of the state values $v$ and the parameters $\theta^\pi$ of the softmax policy $\pi$. Therefore the update $-\alpha\nabla_{\theta^{main}}\mathcal{L}^{RL}$ is the sum of a value update $-\alpha\nabla_{\theta^v}\mathcal{L}^{RL} = \alpha\big(G_t^v - v(x_t)\big)\frac{\partial v(x_t)}{\partial\theta^v}$ and a policy update $-\alpha\nabla_{\theta^\pi}\mathcal{L}^{RL} = \alpha\big(G_t^v - v(x_t)\big)\frac{\partial\log\pi(a_t|x_t)}{\partial\theta^\pi}$, where $G_t^v = (\sum_{j=0}^{j=W}\gamma^j R_{t+j+1}) + \gamma^{W+1}v(x_{t+W+1})$ is a multi-step truncated return, using the agent's estimates $v$ of the state values for bootstrapping after $W$ steps.

The *answer network* parameters $\{\theta^y\}$, instead, are only affected by the *second* term of the update in Equation 2. Since the answers estimate on-policy, under $\pi$, an expected cumulative discounted sum of cumulants, we may use a generalized temporal difference learning algorithm to update $\theta^y$. In our agents, the vector $y$ is a linear function of state, and therefore each GVF prediction $y_i$ is separately parameterised by $\theta^{y_i} \subseteq \theta^y$. The update $-\alpha\nabla_{\theta^{y_i}}\mathcal{L}^{ans}$ for parameters $\theta^y$ may then be written as $\alpha\big(G_t^{y_i} - y_i(x_t)\big)\nabla_{\theta_i^y}y_i(x_t)$, where $G_t^{y_i}$ is the multi-step, truncated, $\gamma_i$-discounted sum of cumulants $u_i$ from time $t$ onwards. As in the main task updates, the notation $G_t^{y_i}$ highlights that we use the answer network's own estimates $y_i(x_t) = x_t^T\theta_i^y$ to bootstrap after a fixed number steps.

The main-task and answer-network pseudo losses $\mathcal{L}^{RL}$, $\mathcal{L}^{ans}$ used in the updates above can also be straightforwardly used to instantiate equation 2 for the parameters $\theta^{enc}$ of the *encoder network*, and to instantiate equation 3, for the parameters $\eta$ of the *question network*. For the shared state representation, $\theta^{enc}$, we explore two updates: (1) using the gradients from both the main task and the answer network, i.e., $-\alpha'\nabla_{\theta_{k-1}}\mathcal{L}^{RL}(\theta_{k-1}) - \alpha'\nabla_{\theta_{k-1}}\mathcal{L}^{ans}(\theta_{k-1})$, and (2) using only the gradients from the answer network, $-\alpha'\nabla_{\theta_{k-1}^{enc}}\mathcal{L}^{ans}(\theta_{k-1})$. Using both the main-task and the answer network components is more consistent with the existing literature on auxiliary tasks, but ignoring the main-task updates provides a more stringent test of whether the algorithm is capable of meta-learning questions that can drive, even on their own, the learning of an adequate state representations.

# 3 Experimental setup

In this section we outline the experimental setup, including the environments we used as test-beds and the high level agent and neural network architectures. We refer to the Appendix for more details.

## 3.1 Domains

*Puddleworld domain:* is a continuous state gridworld domain (Degris et al., 2012), where the state space is a 2-dimensional position in $[0,1]^2$. The agent has 5 actions, where four of these actions move the agent in one of the four cardinal directions by a *mean* offset of $0.05$ and the last action has an offset of $0$. The actions have a stochastic effect on the environment because, on each step, uniform noise sampled in the range $[-0.025, 0.025]$ is added to each action component. We refer to Degris et al. (2012) for further details about this environment.

*Collect-objects domain:* is a four-room gridworld, where the agent is rewarded for collecting two objects in the right order. The agent moves deterministically in one of four cardinal directions. For each episode the starting position is chosen randomly. The locations of the two objects are the same across episodes. The agent receives a reward of $1$ for picking up the first object and a reward of $2$ for picking up the second object after the first one. The maximum length of each episode is $40$.

*Atari domain:* the Atari games were designed to be challenging and fun for human players, and were packaged up into a canonical benchmark for RL agents: the Arcade Learning Environment (Bellemare et al., 2013; Mnih et al., 2015, 2016; Schulman et al., 2015, 2017; Hessel et al., 2018). When summarizing results on this benchmark, we follow the common approach of first normalizing scores on the each game using the scores of random and human agents (van Hasselt et al., 2016).

## 3.2 Our agents

For the gridworld experiments, we implemented meta-gradients on top of a 5-step actor-critic agent with 16 parallel actor threads (Mnih et al., 2016). For the Atari experiments, we used a 20-step IMPALA (Espeholt et al., 2018) agent with 200 distributed actors. In the non-visual domain of

Puddleworld, the encoder is a simple MLP with two fully-connected layers. In other domains the encoder is a convolutional neural network. The main-task value and policy, and the answer network, are all linear functions of the state $x_t$. In the gridworlds the question network outputs a set of cumulants, and the discount factor that jointly defines the GVFs is hand-tuned. In our Atari experiments the question network outputs both the cumulants and the corresponding discounts. In all experiments we report scores and curves averaging results from 3 independent runs of each agent, task or hyperparameter configuration. In Atari we use a single set of hyper-parameters across all games.

### 3.3 Baselines: handcrafted questions as auxiliary tasks

In our experiments we consider the following baseline auxiliary tasks from the literature.

*Reward prediction:* This baseline agent has no question network. Instead it uses the scalar reward obtained at the next time step as the target for the answer network. The auxiliary task loss function for the reward prediction baseline is, $\mathcal{L}^{ans} = \left[y_t(x_t) - r_{t+1}\right]^2$.

*Pixel control:* This baseline also has no question network. The auxiliary task is to learn to optimally control changes in pixel intensities. Specifically, the answer network must estimate optimal action values for cumulants $c_i$ corresponding to the average absolute change in pixel intensities, between consecutive (in time) observations, for each cell $i$ in an $n \times n$ non-overlapping grid overlayed onto the observation. The auxiliary loss function for the action values of the $i^{th}$ cell is: $\mathcal{L}_i^{ans} = \frac{1}{2}\mathbb{E}_{s,a,s'\sim\mathcal{D}}||G_{c_i} + \gamma \max_{a'} q_i^-(s', a') - q_i(s, a)||^2$, where $G_{c_i}$ refers to discounted sum of pseudo-rewards for the $i^{th}$ cell. The auxiliary loss is summed over the entire grid $\mathcal{L}^{ans} = \sum_i \mathcal{L}_i^{ans}$.

*Random questions:* This baseline agent is the same as our meta-gradient based agent except that the question network is kept fixed at its randomly initialized parameters through training. The answer network is still trained to predict values for the cumulants defined by the fixed question network.

## 4 Empirical findings

In this section, we empirically investigate the performance of the proposed algorithm for discovery, as instantiated in Section 2.4. We refer to our meta-learning agent as the "Discovered GVFs" agent. Our experiments address the following questions:

1. Can meta-gradients discover GVF-questions such that learning the answers to them is sufficient, on its own, to build representations good enough for solving complex RL tasks? We refer to these as the "representation learning" experiments.

2. Can meta-gradients discover GVFs questions such that learning to answer these along side the main task improves the data efficiency of an RL agent? In these experiments the representation is shaped by both the updates based on the discovered GVFs as well as the main task updates; we will thus refer to these as the "joint learning" experiments.

3. In both settings, how do auxiliary tasks discovered via meta-gradients compare to handcrafted tasks from the literature? Also, how is performance affected by design decisions such as the number of questions, the number of inner steps used to compute meta-gradients, and the choice between area under the curve versus final loss as meta-objective?

We note that the "representation learning" experiments are a more stringent test of our meta-learning algorithm for discovery, compared to the "joint learning" experiments. However, the latter is consistent with the literature on auxiliary tasks and can be more useful in practice.

### 4.1 Representation learning experiments

In these experiments, the parameters of the encoder network are unaffected by gradients from the main-task updates. Figures 2 and 3 compare the performance of our meta-gradient agents to the baseline agents that train the state representation using the hand-crafted auxiliary tasks described in Section 3.3. We always include a reference curve (in black) corresponding to the baseline actor-critic agent with no answer or question networks, where the representation is trained directly using the main-task updates. We report results for the Collect-objects domain, Puddleworld, and three Atari games (more are reported in the Appendix). From the experiments we highlight the following:

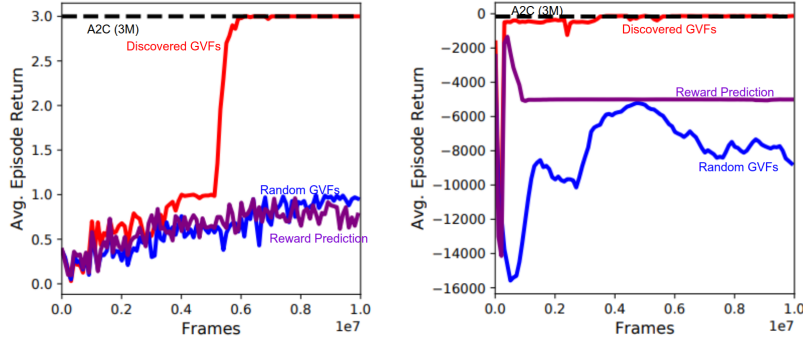

Figure 2: Mean return on Collect-Objects (Left) and Puddleworld (Right) for the "Discovered GVFs" agent (red), alongside the "Random GVFs" (blue) and "Reward Prediction" (purple) baselines. The dashed (black) line is the final performance of an actor-critic whose representation is trained using the main task updates.

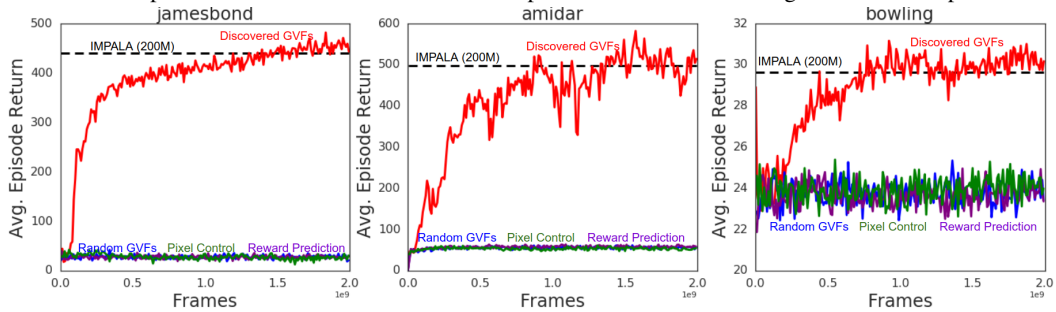

Figure 3: Mean episode return on 3 Atari domains for the "Discovered GVFs" agent (red), alongside the "Random GVFs" (blue), "Reward Prediction" (purple) and "Pixel Control" (green) baselines. The dashed (black) line is the final performance of an actor-critic whose representation is trained with the main task updates.

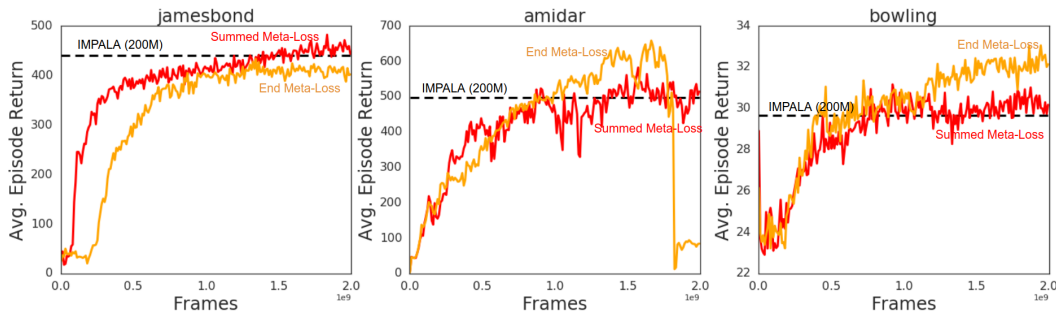

Figure 4: Mean episode return on 3 Atari domains for two "Discovered GVFs" agents optimizing the "Summed Meta-Loss" (red) and the "End Meta-Loss" (Orange), respectively. The dashed (black) line is the final performance of an actor-critic whose representation is trained with the main task updates.

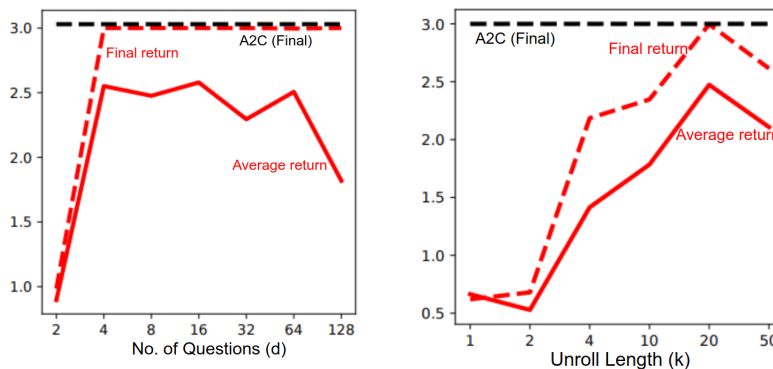

Figure 5: Parameter studies, on Collect-Objects, for "Discovered GVFs" agent, as a function of the number of questions used as auxiliary tasks (on the left) and the number of steps unrolled to compute the meta-gradient (on the right). The dashed and solid red lines correspond to the final and average episode return, respectively.

*Discovery:* in all the domains, we found evidence that the state representation learned solely through learning the GVF-answers to the discovered questions was sufficient to support learning good policies. Specifically, in the two gridworld domains the resulting policies were optimal (see Figure 2); in the Atari domains the resulting policies were comparable to those achieved by the state of the art IMPALA agent after training for 200M frames (see Figure 3). This is one of our main results, as it confirms that non-myopic meta-gradients can *discover* questions, in the forms of cumulants and discounts, *useful* to capture rich enough knowledge of the world to support the learning of state-representations that yield good policies even in complex RL tasks.

*Baselines:* we also found that learning the answers to questions discovered using meta-gradients resulted in state representations that supported better performance, on the main task, compared to the representations resulting from learning the answers to popular hand-crafted questions in the literature. Consider the gridworld experiments in Figure 2, learning the representation using "Reward Prediction" (purple) or "Random GVFs" (blue) resulted in notably worse policies than those learned by the agent with "Discovered GVFs". Similarly, in Atari (shown in Figure 3) the handcrafted auxiliary tasks, now including a "Pixel Control" baseline (green), resulted in almost no learning.

*Main-Task driven representations:* Note that the actor-critic agent that trained the state representation using the main-task updates directly learned faster than the agents where the representation was exclusively trained using auxiliary tasks. The baseline required only 3M steps on the gridworlds and 200M frames on Atari to reach the final performance. This is expected and it is true both for our meta-gradient solution as well as the auxiliary tasks from the literature.

We used the representation learning setting to investigate a number of design choices. First, we compare optimizing the area under the curve over the length of the unrolled meta-gradient computation (or "Summed Meta-Loss") to computing the meta-gradient on the last batch alone ("End Meta-Loss"). As shown in Figure 4, both approaches can be effective, but we found that optimizing area under the curve to be more stable. Next we examined the role of the number of GVF questions, and the effect of varying the number of steps unrolled in the meta-gradient calculation. For this purpose, we used the less compute-intensive gridworlds: Collect-Objects (reported here) and Puddleworld (in the Appendix). On the left in Figure 5, we report a parameter study, plotting the performance of the agent with meta-learned auxiliary tasks as a function of the number of questions $d$. The dashed black line corresponds to the optimal (final) performance. Too few questions ($d = 2$) did not provide enough signal to learn good representations: the dashed red line is thus far from optimal for $d = 2$. Other values of $d$ all led to learning of a good representation capable of supporting an optimal policy. However, too many questions (e.g. $d = 128$) made learning slower, as shown by the average performance dropping. The number of questions is therefore an important hyperparameter of the algorithm. On the right, in Figure 5 we report the effect on performance of the number $k$ of unrolled steps used for the meta-gradient computation. Using $k = 1$ corresponds to the myopic meta-gradient: in contrast to previous work (Xu et al. (2018); Zheng et al. (2018)), the representation learned with $k = 1$ and $k = 2$ was insufficient for the final policy to do anything meaningful. Performance generally got better as we increased the unroll length (although the computational cost of meta-gradients also increased). Again the trend was not fully monotonic, with the largest unroll length $k = 50$ performing worse than $k = 25$ both in terms of final and average performance. We conjecture this may be due to the increased variance of the meta-gradient estimates as the unroll length increases. The number of unrolled steps $k$ is therefore also a sensitive hyperparameter. Note that neither $d$ nor $k$ were tuned in other experiments, with all other results using the same fixed settings of $d = 128$ and $k = 10$.

## 4.2 Joint learning Experiments

The next set of experiments use the most common setting in the literature on auxiliary tasks, where the representation is learned using *jointly* the auxiliary task updates and the main task updates. To accelerate the learning of useful questions, we provided the encoded state representation as input to the question network instead of learning a separate encoding; this differs from the previous experiments, where the question network was a completely independent network (consistently with the objective of a more stringent evaluation of our algorithm). We used a benchmark consisting of 57 distinct Atari games to evaluate the "Discovered GVFs" agent together with an actor-critic baseline ("IMPALA") and two auxiliary tasks from the literature: "Reward Prediction" and "Pixel Control".

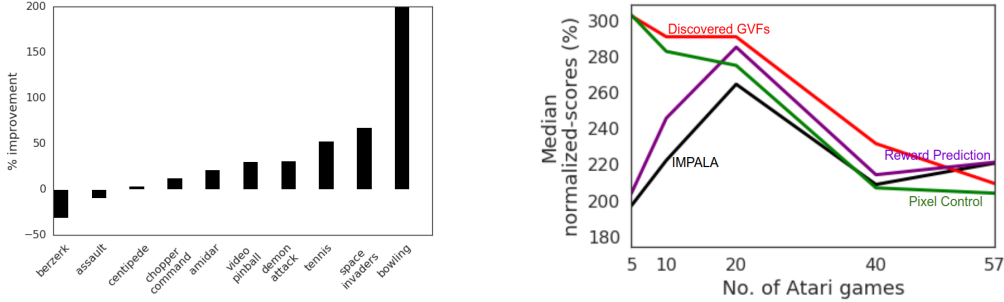

Figure 6: On the left, relative performance improvements of a "Discovered GVF" agent, over plain IMPALA. The 10 games are those where a "Pixel Control" baseline showed the largest gains over IMPALA. On the right, we plot median normalized scores of all agents for different subsets of the 57 Atari games (N=5, 10, 20, 40, 57). The order of inclusion of the games is again determined according to the performance gains of pixel-control.

None of the auxiliary tasks outperformed IMPALA on each and every of the 57 games. To analyse the results, we ranked games according to the performance of the agent with pixel-control questions, to identify the games more conducive to improving performance through the use of auxiliary tasks. On the left of Figure 6, we report the relative gains of the "Discovered GVFs" agent over IMPALA, on the top-10 games for the "Pixel Control" baseline: we observed large gains in 6 out of 10 games, small gains in 2, and losses in 2. On the right in Figure 6, we provid a more comprehensive view of the performance of the agents. For each number $N$ on the x-axis ($N = 5, 10, 20, 40, 57$) we present the median human normalized score achieved by each method on the top-$N$ games, again selected according to the "Pixel Control" baseline. It is visually clear that discovering questions via meta-learning is fast enough to compete with handcrafted questions, and that, in games well suited to auxiliary tasks, it greatly improved performance over all baselines. It was particularly impressive to find that the meta-gradient solution outperformed pixel control on these games despite the ranking of games being biased in favour of pixel-control. The reward prediction baseline is interesting, in comparison, because it's profile was the closest to that of the actor-critic baseline, never improving performance significantly, but not hurting either.

## 5    Conclusions and Discussion

There are many forms of questions that an intelligent agent may want to discover. In this paper we introduced a novel and efficient multi-step meta-gradient procedure for the discovery of questions in the form of on-policy GVFs. In a stringent test, our representation learning experiments demonstrated that the meta-gradient approach is capable of discovering useful questions such that answering them can drive, by itself, learning of state representations good enough to support the learning of a main reinforcement learning task. Furthermore, our auxiliary tasks experiments demonstrated that the meta-learning based discovery approach is data-efficient enough to compete well in terms of performance, and in many cases even outperform, handcrafted questions developed in prior work.

Most prior work on auxiliary tasks relied on human ingenuity to define questions useful for shaping the state representation used in a given task, but it's hard to create questions that are both useful and general (i.e., that can be applied across many tasks). Bellemare et al. (2019) introduced a geometrical perspective to understand when auxiliary tasks give rise to good representations. Our solution differs from this line of work in that we side-step the question of how to *design* good auxiliary questions, by meta-learning them instead, directly optimizing for utility in the context of a given task. Our approach fits in a general trend of increasingly relying on data rather than human designed inductive biases to construct effective learning algorithms (Silver et al., 2017; Hessel et al., 2019b).

A promising direction for future research is to investigate *off-policy* GVFs, where the policy under which we make the predictions differs from the main-task policy. We also note that our approach to discovery is quite general, and could be extended to meta-learning other kind of questions, that do not fit the canonical GVF formulation; see van Hasselt et al. (2019) for one such class of predictive questions. Finally, we emphasize that the unrolled multi-step meta-gradient algorithm is likely to benefit both previous applications of myopic meta-gradients, as well as possibly open up more applications, other from discovery, where the myopic approximation would fail.

**Acknowledgments**

We thank John Holler and Zeyu Zheng for many useful comments and discussions. The work of the authors at the University of Michigan was supported by a grant from DARPAs L2M program and by NSF grant IIS-1526059. Any opinions, findings, conclusions, or recommendations expressed here are those of the authors and do not necessarily reflect the views of the sponsors.

## Footnotes

[1]University of Michigan, Ann Arbor. Corresponding author: Vivek Veeriah ⟨vveeriah@umich.edu⟩

[2]DeepMind, London.

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
