[Supplementary Material · Discovery of Useful Questions as Auxiliary Tasks - NeurIPS 2019 Camera Ready Appendix.pdf]

# 6 Appendix

## 6.1 Neural network architecture and details

**Representation learning experiments:**

*Puddleworld domain:* A multi-layer perceptron (MLP), with layer fully-connected layers with $128$ hidden units each. ReLU activation functions are used throughout.

*Collect-Objects domain:* A two-layer convolutional neural network (CNN) with $8, 16$ filters in each layer respectively. The filter sizes were $2 \times 2$ in both layers. The CNN's output is then fed to a fully-connected layer with $512$ hidden units. ReLU activation functions are used throughout.

*Atari domain:* A three-layer CNN architecture that has been successfully used on Atari in several variants of DQN (Mnih et al., 2015; van Hasselt et al., 2016; Hessel et al., 2018). The CNN layers consists of $32, 64, 64$ filters respectively, with filter sizes $8 \times 8, 4 \times 4, 3 \times 3$. The stride lengths at each of these layers were set to $4, 2, 1$ respectively. ReLU activation functions are used throughout.

In all cases, the output of the encoding modules are linearly mapped to produce the policy, value function and answer net heads. ReLU activations are used in the learning agent.

We use an independent question network in all representation learning experiments; the architecture of the hidden layers matches that of the learning agent exactly. Note, however, that the heads of the question network output cumulants and discounts to be used as questions, and these are both vectors, of the same size $D = 128$. We use $arctan$ activations for cumulants and a $sigmoid$ for discounts.

**Joint learning experiments:**

*Atari domain:* We use a Deep ResNet architecture identical to the one from Espeholt et al. (2018). They only differ in the outputs: as we now have an answer head, in addition to policy and values.

The question net takes the last hidden layer of the ResNet as input; it uses a meta-trained two-layer MLP to produce cumulants, and a separately parameterised two-layer MLP to produce discounts. The MLPs have $256, 128$ hidden units respectively, with ReLU activations in both. As in the representation learning experiments, we use $arctan$ activations for cumulants and a $sigmoid$ for discounts.

## 6.2 Hyperparameters used in our experiments

**Representation learning experiments:**

*A2C:* The A2C agents (used in the gridworld domains) use 5-step returns in the $\mathcal{L}^{RL}$ pseudo-loss. We searched the initial learning rate for the RMSProp optimizer and the entropy regularization coefficient in a range of values, and the best combination of these hyperparameter was chosen according to the results of the A2C baseline, and then used for all agents. The range of values for the initial learning rate hyperparameter was: $\{0.0001, 0.0003, 0.0007, 0.001, 0.003, 0.007, 0.001\}$. The range of values for entropy regularization was: $\{0.0001, 0.001, 0.01, 0.03, 0.05\}$. The hyperparameter $\epsilon$ of the RMSProp optimizer is set to $1 \times 10^{-05}$. The number of unrolling steps $k$ is set to $k = 10$.

*IMPALA:* All agents based on IMPALA (used in Atari domains) uses the hyperparameters reported by Espeholt et al. (2018). They are listed in Table 1 together with the hyper-parameters specific to DGVF and to other baselines. The number of unrolling steps for meta-gradients is $k = 10$ .

**Joint learning experiments:** The hyperparameters specific to the auxiliary tasks are obtained by a search over ten games (ChopperCommand, Breakout, Seaquest, SpaceInvaders, KungFuMaster, MsPacman, Krull, Tutankham, BattleZone, BeamRider) following common practice in Deep RL Atari experiments (Mnih et al., 2015; van Hasselt et al., 2016; Hessel et al., 2018). After choosing the hyperparameter from this search, they remain fixed across all Atari games.

## 6.3 Preprocessing

In the Atari domain, the input to the learning agent consists of $4$ consecutively stacked frames where each frame is a result of repeating the previous action for $4$ time-steps, greyscaling and downsam-

| IMPALA | Value |
| --- | --- |
| Network Architecture | Deep ResNet |
| $n$-step return | 20 |
| Batch size | 32 |
| Value loss coefficient | 0.5 |
| Entropy coefficient | 0.01 |
| Learning rate | 0.0006 |
| RMSProp momentum | 0.0 |
| RMSProp decay | 0.99 |
| RMSProp $\epsilon$ | 0.1 |
| Global gradient norm clip | 40 |
| Learning rate schedule | Anneal linearly to 0 |
| Number of learners | 1 |
| Number of actors | 200 |

| GVF Questions | Value |
| --- | --- |
| Meta learning rate | 0.0006 |
| Meta optimiser | ADAM |
| Unroll length | 10 |
| Meta gradient norm clip (cumulants) | 1 |
| Meta gradient norm clip (discounts) | 10 |
| Number of Questions | 128 |
| Auxiliary loss coefficient | 0.0001 |

| Pixel-Control | Value |
| --- | --- |
| Auxiliary loss coefficient | 0.0001 |

| Reward-Prediction | Value |
| --- | --- |
| Auxiliary loss coefficient | 0.001 |

Table 1: Detailed hyperparameters used by all learning agents based on IMPALA.

pling the resulting frames to 84x84 images, and max-pooling the last 2. This is a fairly canonical pre-processing pipeline for Atari. Additionally rewards are clipped to the [-1, 1] range.

## 6.4 Derivation of myopic approximation to meta-gradients

Here we derive the myopic approximation for our meta-gradient procedure that was previously described in the main text.

$$\frac{\partial \theta_t^{enc}}{\partial \eta} = \frac{\partial}{\partial \eta}\left[\theta_t^{enc} - \alpha\big(y(o_{t-i+1:t}) - u(o_{t+1:t+j})\big)\frac{\partial y(o_{t-i+1:t})}{\partial \theta^{enc}}\right] \tag{4}$$

$$\approx -\frac{\partial}{\partial \eta}\left[\alpha\big(y(o_{t-i+1:t}) - u(o_{t+1:t+j})\big)\frac{\partial y(o_{t-i+1:t})}{\partial \theta^{enc}}\right] \tag{5}$$

$$= \alpha\frac{\partial u(o_{t+1:t+j})}{\partial \eta}\frac{\partial y(o_{t-i+1:t})}{\partial \theta^{enc}}\right] \tag{6}$$

$$\frac{\partial \theta_{t+1}^{\pi}}{\partial \eta} = \frac{\partial}{\partial \eta}\left[\theta_t^{\pi} + \alpha\big(R - V(x_t)\big)\frac{\partial \log \pi(a_t|x_t)}{\partial \theta^{\pi}}\right] \tag{7}$$

$$\approx \frac{\partial}{\partial \eta}\left[\alpha\big(R - V(x_t)\big)\frac{\partial \log \pi(a_t|x_t)}{\partial \theta^{\pi}}\right] \tag{8}$$

$$\frac{\partial \theta_{t+1}^{v}}{\partial \eta} = \frac{\partial}{\partial \eta}\left[\theta_t^{v} + \alpha\beta\big(R - V(x_t)\big)\frac{\partial V(x_t)}{\partial \theta^{v}}\right] \tag{9}$$

$$\approx \frac{\partial}{\partial \eta}\left[\alpha\beta\big(R - V(x_t)\big)\frac{\partial V(x_t)}{\partial \theta^{v}}\right] \tag{10}$$

Equations 5, 8 and 10 are a **myopic** approximation because they ignore the fact that $\theta_\bullet$ is affected by the changes in $\eta$. Furthermore, in Equations 8 and 10, the policy $\pi_{enc,\pi}$ and value function $V_{enc,v}$ are only indirect functions of $\eta$ (i.e., they are indirectly affected by the auxiliary loss) and thus they do not participate in the myopic approximation. Therefore, after applying all the approximations, we get the following myopic update rule for the meta-parameters $\eta$:

$$\eta_{t+1} = \eta_t - \alpha\frac{\partial \mathcal{L}^{a2c}}{\partial \theta^{enc}}\frac{\partial u_\eta(o_{t+1:t+j})}{\partial \eta}\frac{\partial y(o_{t-i+1:t})}{\partial \theta^{enc}}. \tag{11}$$

## 6.5 Comparison between myopic and unrolled meta-gradient

Figure 7 visualizes the computation graph that is a consequence of the unrolled computation for the meta-gradient and the myopic meta-gradient computation. In the unrolled computation, the gradient of the meta-objective w.r.to the meta-parameters ($\eta$) is computed in such a way that the effect of these parameters over a longer time-scale is taken into consideration. The gradient computation for this unrolled computation is given in Equation 4. In contrast, the myopic gradient computation only considers the immediate one time-step effect of the meta-parameters in the agent's policy. The meta-gradient update based on this myopic gradient computation is given in Equation 11.

Figure 7: On the left, the unrolled compute graph that allows efficient computation of the meta-gradient. On the right, the myopic or 1-step version corresponding to the meta-gradients update used in previous work.

## 6.6 Additional Results

*Representation learning experiments:* The aim of the representation learning is to evaluate how well auxiliary tasks can drive, on their own, representation learning in support of a main reinforcement learning task. In Figure 8 we report additional representation learning results for 6 Atari games (jamesbond, gravitar, frostbite, amidar, bowling and chopper command) including the games from the main text. The "Discovered GVFs" (red), "Pixel Control" (green), "Reward Prediction" (purple) and "Random GVFs" (blue) baseline agents all rely exclusively on auxiliary tasks to drive representation learning, while the linear policy and value functions are trained using the main-task updates. In all games the "Discovered GVFs" agent significantly outperforms the baselines using the hand-crafted auxiliary tasks from the literature to train the representation. In two games (gravitar and frostbite) the "Discovered GVFs" significantly outperforms also the plain "IMPALA" agent (trained for 200M frames) that uses the main task updates to train the state representation. In Figure 9 we report the parameter studies for the "Discovered GVFs" agent, in the second gridworld domain Puddleworld; the plots show performance as a function of the number of questions used as auxiliary tasks (on the left) and the number of steps unrolled to compute the meta-gradient (on the right). Again results are consistent with those reported in the main text for the Collect Objects domain.

*Joint learning experiments:* In Figures 11, 12 and 10 we provide additional details for the "joint learning" experiments. The aim of these experiments is to show that whether the process of discovery of useful questions via meta-gradients is fast enough to improve the data efficiency of an agent in a standard setting where the state representation is trained using both the auxiliary task updates as well as the main task updates. We report relative performance improvements achieved by the "Discovered GVFs" agent over the "IMPALA", "Pixel Control" and "Reward Prediction" agents, after 200M training frames, on each of the 57 Atari games. The same hyperparameters are used for all games. The relative improvements are computed using the human normalized final performance of each agent, averaged across 3 replicas of each experiment (for reproducibility).

Figure 8: Mean episode return for several learning agents on 6 different Atari games, including the three games reported in the main text. The solid horizontal line represents the final performance after 200M frames of training of a plain "IMPALA" agent. The "Discovered GVFs" (red), "Pixel Control" (green), "Reward Prediction" (purple) and "Random GVFs" (blue) baseline agents all rely exclusively on auxiliary tasks to drive representation learning, while the linear policy and value functions are trained using the main-task updates.

Figure 9: Parameter studies, on Puddleworld, for the "Discovered GVFs" agent, as a function of the number of questions used as auxiliary tasks (on the left) and the number of steps unrolled to compute the meta-gradient (on the right). The dashed and solid red lines correspond to the final and average episode return, respectively.

Figure 10: Improvement in human normalized performance at the end of training for the "Discovered GVFs" agent, with respect to a "Pixel Control" baseline agent, on each of 57 Atari games. Both are trained for 200M frames.

Figure 11: Improvement in human normalized performance at the end of training for the "Discovered GVFs" agent, with respect to a plain "IMPALA" baseline agent, on each of 57 Atari games. Both are trained for 200M frames.

Figure 12: Improvement in human normalized performance at the end of training for the "Discovered GVFs" agent, with respect to a "Reward Prediction" baseline agent, on each of 57 Atari games. Both are trained for 200M frames.