[Reviews · NeurIPS 2019]

Reviewer 1



In light of the new experiments which mostly validate the approaches capability of improving performance on the main task - I have increased my score from a 6 to a 7. It would have been nice if the new results considered more than one random seed (had confidence intervals etc...). ----------------------------------------------------------------------------------------------------- Originality The paper is well framed, the authors adequately cite related work on GVFs, auxiliary tasks, and meta gradients. Combining meta-gradients and GVFs for auxiliary task discovery is fairly novel. The closest related work used a myopic objective which the authors improve upon with their multi-step version. Quality The multi-step meta-gradient is a simple yet effective way (as the authors show in the experiments) to improve the learning of a representation. On that note, it would have been nice to have a similar ablation of the multi-step objective on more complicated domains such as pacman. The authors are direct in acknowledging some of the limitations of the approach, as it requires maintaining k copies (where k is the unroll length) of the network parameters. Perhaps some additional discussion about whether or not this is harmful in practice (running out of gpu memory) would be useful. Unfortunately, based on the current set of experiments I’m not entirely convinced that there is a concrete scenario where you would actually want / need the proposed method. Since the ultimate goal is to perform better (either in terms of sample efficiency, final performance, or perhaps generalization since their method could lead to a better from generalizable representation) it is striking that the baseline A2C agent outperforms the proposed method across experiments. On that note, not allowing the gradients from the main task to flow to the representation layers is definitely one way to isolate the contributions of the auxiliary tasks. However, it feels more like an ablation study then a main result. I think it is critical that the authors add some experiments where they allow the gradients to flow from the main task and highlight gains with respect to some metric (whether that be sample efficiency, generalization, etc…). Clarity The paper is a very clearly written and well polished paper. All the necessary hyperparameters and experimental setup is provided in either the main text or the supplementary. Significance Overall, the automatic discovery of auxiliary tasks through the proposed multi-step objective would be of great interest to the Deep RL community and despite not necessarily achieving any state of the art results (which would have put this paper over the top) is still an interesting contribution. Small notes: Figures 7/8 in appendix - colors in legend don’t line up (e.g blue doesn’t appear in the legend but does in the figure) Regarding the multi-step meta gradient, I’m wondering if a td lambda style version of this would perform better, where you put more weight to short term losses (since there’s variance in using distant losses). Optimizing hyperparameters for A2C and then using that set of hyperparameters for all other algorithms is obviously not the ideal setup (the ideal setup being doing an individual search for each method) but understandable under hardware constraints. This could explain to some degree why the A2C baseline is the best performing agent in all experiments.

Reviewer 2



Summary: The paper introduces a SGD based learner for the discovery of questions in the form of General Value Functions (GVFs) [Sutton et al. 2011] as auxiliary tasks. A question network is trained to predict real values based on previous observations such that an answer network has to predict the discounted cumulative value of these real values. A gradient of a gradient (meta-gradient) is used to update the parameters of the question network to maximize the cumulative return of the policy that relies on the weights of the answer network. Significance In principle, auxiliary predictive tasks may be able to speed up training or improve the final policy quality when used in conjunction with a standard RL objective. In the reviewed paper, this has not been shown, and I am not confident that the presented results are extremely relevant. Furthermore, there are other issues in terms of clarity and empirical evaluation. Clarity / Quality The introduction lacks clarity. While general value functions are central to the paper, their correspondence to predictive questions is not formally defined and the reader is required to be familiar with the concept. The paper is poorly motivated. Why are such predictive questions important? Some claims are not given evidence for or are imprecise, e.g. line 23, 'GVFs are very expressive' and line 37, 'Our second contribution is a precise and efficient algorithm' Originality and missing related work The author's second contribution of doing multiple meta-gradient steps has appeared in similar form in MAML [Finn et al. 2017] (and possibly even before that) and I would not consider it highly novel. l 16-21 and 55-61: the authors mention recent work on auxiliary tasks, but not the original work on inventing explicit, well-defined auxiliary tasks or questions or problems, namely, the PowerPlay framework of 2011 [1,2]. PowerPlay automatically creates a sequence of auxiliary tasks/skills for Reinforcement Learning (RL), with a bias towards simple tasks, including action and prediction tasks, including predictions of internal states as consequences of action sequences, etc. PowerPlay can perceive and generate descriptions of arbitrary tasks with computable solutions. It learns one new computable task after another and uses environment-independent replay of behavioral traces (or functionally equivalent but more efficient approaches) to avoid forgetting previous skills. Refs [2,3] have less strict variants of PowerPlay. Of course, the early approaches to RL with curiosity since 1990 [4-6] also defined auxiliary tasks through intrinsic rewards, so that's even older, but back then the tasks were not clearly separated and formally defined through goal-defining extra inputs, like in PowerPlay [1,2]. One should probably point out this difference. Given the above, the authors should clarify what's really novel in their approach. Here a list of relevant references from Schmidhuber's lab, with various authors: [1] POWERPLAY: Training an Increasingly General Problem Solver by Continually Searching for the Simplest Still Unsolvable Problem. Frontiers in Psychology, 2013. Based on arXiv:1112.5309v1 [cs.AI], 2011. (Very general framework for automatically defining a curriculum of incremental learning tasks.) [2] First experiments with PowerPlay. Neural Networks, 41(0):130-136, Special Issue on Autonomous Learning, 2013. [3] Continual curiosity-driven skill acquisition from high-dimensional video inputs for humanoid robots. Artificial Intelligence, 2015. (Intrinsic motivation from raw pixel vision for a real robot.) [4] A possibility for implementing curiosity and boredom in model-building neural controllers. In Proc. SAB'91, pages 222-227. MIT Press/Bradford Books, 1991. Based on [8]. [5] Making the world differentiable: On using fully recurrent self-supervised neural networks for dynamic reinforcement learning and planning in non-stationary environments. TR FKI-126-90, TU Munich, 1990. http://people.idsia.ch/~juergen/FKI-126-90_(revised)bw_ocr.pdf [6] Curious model-building control systems. In Proc. International Joint Conference on Neural Networks, Singapore, volume 2, pages 1458-1463. IEEE, 1991. Methodological / Evaluation Why would one artificially constraint the setup to not backpropagate the error from the RL algorithm into the learned representation [where the learned representations are updated through the learned general value function]? I agree that, as the authors stated, this does help to assess whether the auxiliary rewards can learn useful representations. But it is unclear to me why these learned representations are useful at all if they don't ultimately lead to an algorithm that is more sample efficient or reaches better final cumulative reward compared to the base RL algorithm (here A2C). Instead, transfer to other environments / agents would be interesting, i.e. how well the learned general value functions generalize. Alternatively, the use of auxiliary tasks could speed up training / improve final cumulative reward in complex environments. This, however, was not shown in the present paper. Furthermore, some baselines would have been interesting: A linear policy network trained by A2C. This should be the lower bound for any method because it would correspond to an encoder that just passes the observations through. Random encoder network (Instead of GVF system). Random projections of the observations might also aid training of a linear policy and should be outperformed by a learned system. Figure 2 is missing the pixel control and value prediction baselines, although these appear in the legend. For now, I vote for rejecting this submission, although I would not be upset if it were accepted, provided the comments above were addressed in a satisfactory way - let us wait for the rebuttal! Response to rebuttal: Provided the revised version (which we would like to see again) takes into account all comments as promised in the rebuttal, we increase our score from 4 to 6! Here our thoughts on how the authors have addressed each of our concerns: 1. The authors promise to revise their related work section as requested. 2. "Why would one artificially constraint the setup to not backpropagate the error from the RL algorithm into the learned representation [where the learned representations are updated through the learned general value function]?" This has been addressed with new results in the rebuttal l 1-10 3. "The paper is poorly motivated. Why are such predictive questions important? It is unclear why these learned representations are useful at all if they don't ultimately lead to an algorithm that is more sample efficient or reaches better final cumulative reward compared to the base RL algorithm (here A2C)." The new results in figure 1 suggest that indeed the learned auxiliary losses are improving final cumulative rewards compared to the A2C baseline. 3. "In principle, auxiliary predictive tasks may be able to speed up training or improve the final policy quality when used in conjunction with a standard RL objective. In the reviewed paper, this has not been shown, and I am not confident that the presented results are extremely relevant." See above Conclusion: These new results considerably boost the paper and address most of our concerns. While the paper could be further improved through multiple seeds, confidence intervals for all plots, and a wider range of environments and RL algorithms, the presented auxiliary task discovery in the form of GVFs now seems to be of adequate novelty with supporting results.

Reviewer 3



Originality: As far as I am aware, using neural networks to define generalized value functions that are to be learned as an auxiliary task is a novel and interesting combination of existing ideas, and could be used in future work. More in general, the automatic adaptation of auxiliary tasks has been previously hinted at, but not explored in depth; in that regard too, the authors present work that is new to the field. I do feel, however, that the novelty of the meta-gradient computation procedure is being overstated, as it seems equivalent to [1]. Could the authors kindly explain the difference between the two algorithms? Quality: Overall the paper is technically sound (sections 2 and 3), though with a few small issues that I would like to authors to comment on. Line 94: the authors state that the backwards pass can be computed at a cost similar to that of the forward computation. While generally true, I find this statement a bit misleading. If my understanding is correct, this method computes several forward (and thus backward) passes per single meta-update, which requires significant computation. Line 285: the authors claim that the “variance of the meta-gradient estimates increases with the unroll length”. I’m not sure how this conclusion can be made from the graphs in the supplementary material, which only show that increasing the rollout length leads to an increased variance *of the returns*. The proposed method sounds interesting and promising to me. However, I am not convinced by the experimental evaluation (this is my main criticism of the paper). The experiments demonstrate that the meta-learning procedure is capable of improving the usefulness of the auxiliary tasks when these are the only source of representation learning signal, but this scenario is somewhat contrived (usually the main task also contributes to representation learning). No experiments explored how this algorithm performs when gradients are allowed to flow from the main A2C task to the encoder. I wonder therefore whether it might be possible that the meta-learning procedure actually simply amounts to minimizing the A2C loss w.r.t. the encoder through a different (longer and partially implicit) computation graph. In that case the meta-learning procedure would not provide the encoder with any information that it could not receive from the A2C loss, were that being “directly” minimized w.r.t. the encoder. Then one could expect none of the meta-RL agents to perform better than the A2C baseline, which is what we see in the experimental results. Clarity: The quality of writing of the paper is good overall. In particular, the authors explain the proposed algorithm and the environments used to evaluate it thoroughly and with clarity. As a result, an expert reader with access to the supplementary material would be able to reproduce the results without significant obstacles. However, I found parts of the paper difficult to follow, since some key concepts such as generalized value functions were not clearly defined, and the review of related literature was somewhat scarce. For example, the section relevant to meta-gradients focused on only two specific examples from reinforcement learning, and did not consider work on meta-gradients beyond RL. I also found the terminology of “questions” and “answers” a bit confusing, as it is not commonly used in the multi-task literature. Significance: Auxiliary task selection and/or generation in asymmetric multi-task learning is an important yet relatively unexplored issue. Indeed, current practice is to design auxiliary tasks manually, and as such, work in this area has the potential of being highly significant. Unfortunately, in its current form the paper falls short of demonstrating that the proposed algorithm is a viable replacement for manual auxiliary task design. It can however be seen as an encouraging exploratory result and as the basis for interesting further investigation, which I warmly encourage the authors to pursue. References: [1] Marcin Andrychowicz, Misha Denil, Sergio Gomez, Matthew W Hoffman, David Pfau, Tom Schaul, Brendan Shillingford, and Nando De Freitas. Learning to learn by gradient descent by gradient descent. In Advances in neural information processing systems, pages 3981–3989, 2016. ---------------------------------------------------------------------------------------------------- -- UPDATE -- I read the author's feedback and other reviews. I was happy to see that the authors addressed most of the concerns initially raised, and the new results look promising. I therefore increase the score of my review from 5 to 6. I still have some remaining concerns which I hope the authors will consider for the next revision of their paper: - It seems like the main difference between the author's method and L2L is that they apply it to RL instead of supervised learning. Therefore the meta-update computation is not itself a novel contribution, which should be clarified in the paper. - I think that the most significant contribution made by the authors is showing that their method could be used to generate auxiliary tasks that are more helpful than hand-engineered ones. I believe that the authors should clearly focus on this aspect of their research. The new results should be confirmed by testing the method on more seeds and a broader range of environments.

[Author Response · NeurIPS 2019]

**Response to all reviewers:** We have significantly updated the results of the paper to show that our meta-gradient
method can indeed learn auxiliary questions fast enough to improve learning performance on the main task. In these
experiments, as suggested, we use both the actor-critic loss and the (continuously adapting) meta-learned auxiliary
question losses to update the state representation, as is usually done in work using hand-crafted auxiliary losses. We
compared our approach to a baseline agent that uses only the actor-critic loss, and to agents using both the actor-critic
loss and two hand-crafted auxiliary losses introduced in UNREAL: reward-prediction, and pixel-control. Our approach
performs significantly better on 7 games, and performs significantly worse on just 1 game of the 10 Atari games tried
so far (see Figure 1). This suggests that discovering GVF questions using meta-gradients is a promising approach for
auxiliary task discovery that improves learning. As in UNREAL, both our agents and the corresponding baselines were
trained for 200M frames, and hyperparameters such as the weighting of the auxiliary loss were tuned per game.

Figure 1: *New results:* **Performance improvement of an agent trained with both actor-critic and (continu-
ously adapting) meta-gradient-learned auxiliary question losses, compared to three baseline agents after 200M
frames of training.** The baselines are the standard actor-critic agent, and agents trained with both the actor-critic loss
and one of two hand-crafted auxiliary losses: reward-prediction and pixel-control. Our approach performs significantly
better on 7 Atari games while performing significantly worse on 1 Atari game of the 10 tried so far.

(**R1**): We extended our representation-learning experiments to include more complex Atari games; as in the experiments
in the submitted paper, here only the meta-learned auxiliary question loss is used to update the state representation.
Figure 2 shows performance on 5 such games. As in the results in the submitted paper, the meta-gradient approach can
discover questions in the form of GVFs that drive the learning of a state representation that is good enough to support
the main-task learning, even on these more challenging RL domains. Note that this is not the case for the baseline
auxiliary tasks (pixel-control and reward-prediction).

(**R2** & **R3**): We agree that there is a strong relationship between meta-gradient and MAML/L2L. However, we highlight
several important differences: a) meta-gradient RL updates meta-parameters of the update function (in this case the
auxiliary tasks), whereas MAML (typically) updates the initial parameters, b) meta-gradient RL may be applied to
adapt and improve performance during the lifetime of a single agent, whereas MAML learns across many lifetimes
(impractical in many applications), c) Our paper demonstrates this idea on far more complex RL tasks (needing millions
of training steps as opposed to hundreds). We will revise our claim by stating these additional details and add the
necessary citations to clarify this point. Thanks. We do prefer to continue using the questions/answer terminology to
highlight the conceptual similarity to Temporal-Difference network, but agree that we should more carefully introduce
and motivate both the terminology and the question framing in the revision.

(**R2**): We thank the reviewer for highlighting interesting related work: PowerPlay is indeed similarly motivated as our
submission, but uses greedy search over all possible task descriptions for discovering tasks instead of meta-gradients.
Curiosity-based RL uses hand-designed intrinsic rewards. We will clarify the relation to our method in our revision and
emphasize that our primary contribution is the discovery of GVFs as auxiliary questions through meta-gradients (as
pointed out by **R1**).

(**R3**): The algorithm does perform several forward passes per meta-update; we will clarify this in the revision. We will
also clarify that performance decreases if the unroll length is increased too much, and we will perform new experiments
to confirm the hypothesis that variance increases with unroll length. Thanks for pointing this out.

Figure 2: *New results*: **Additional more complex Atari tasks to test representation learning driven only by the
meta-learned auxiliary questions, compared to hand-crafted baselines.** The plot includes a horizontal black line
that shows the episode return achieved by a standard actor-critic agent after 200M training frames along with the learning
curves of the meta-gradient discovery approach, and hand-crafted reward-prediction and pixel-control auxiliary losses.
We demonstrate that our approach of discovering auxiliary questions in the form of GVFs can drive representation
learning which eventually allows the agent to reach a comparable level of performance to the standard actor-critic agent.
These new results show that the approach scales to complex RL domains.

[Meta-Review · NeurIPS 2019]

There is consensus about the interest of the algorithm and the empirical results. Yet, the reviewers pointed out several dimensions along which the current submission can be improved: - Clarify relationship with L2L and the novelty of the proposed algorithm - Include more complete empirical validation along the lines of what reported in the rebuttal - Expand the related work discussion I encourage the authors to take these recommendations into account in preparing the final version of the paper.